# Ultra-high electrostriction and ferroelectricity in poly (vinylidene fluoride) by 'printing of charge' throughout the film

Ningyi Zhang[1,2] ✉, Xiaobing Dong[1,2], Shihui He[1,2], Zhao Liang[1,3], Weipeng Li[1,2], Qihao Qian[1,2] & Chao Jiang ●[1,2] ✉

Electrostriction is an important electro-mechanical property in poly (vinylidene fluoride) (PVDF) films, which describes the proportional relation between the electro-stimulated deformation and the square of the electric field. Generally, traditional methods to improve the electrostriction of PVDF either sacrifice other crystalline-related key properties or only influence minimal regions around the surface. Here, we design a unique electret structure to fully exploit the benefits of internal crystal in PVDF films. Through the 3D printing of charged ink, we have obtained the best electrostrictive and ferroelectric properties among PVDF-based materials so far. The optimized electrostrictive coefficient $M_{33}$ ($324 \times 10^{-18}$ m$^2$ V$^{-2}$) is $10^4$ times that of normal PVDF films, and the piezoelectric constant $d_{33}$ (298 pm V$^{-1}$) is close to 10 times its traditional limit. The proposed 3D electret structure and the bottom-up approach to 'print the charge' open up a new way to design and adapt the electroactive polymers in smart devices and systems.

Due to the excellent dielectric, ferroelectric, pyroelectric, and other electro-mechanical (EM) properties, poly (vinylidene fluoride) (PVDF) and its copolymers show a wide range of applications in actuators, sensors, energy harvesters, and other fields[1,2]. In recent years, they become more attractive as the core component of the flexible and wearable devices, used in biomedical engineering[3,4], new energy applications[5,6], and artificial intelligent systems[7]. Since these properties are generally believed to originate from the polar crystal phases[8], optimizing the crystal phase becomes the key to improving PVDF's ferroelectric and other EM properties[9,10]. In fact, the improvement of ferroelectric property in PVDF is generally based on enhancing the composition and polarization of its $\beta$ phase[2,11]. However, as an essential EM property of PVDF, electrostriction can hardly be optimized by adjusting crystal polar phases. In the past, the electrostrictive property of PVDF has long been considered an intrinsic property that is mainly susceptible to its carbon chain structure[12,13]. Therefore, much research has significantly enhanced the electrostrictive performance of PVDF by

changing their carbon chain structure or degree of cross-linking, such as physical transition[14,15] and chemical reaction[16]. Unfortunately, these methods impact the original crystal structure, sharply weakening the ferroelectric properties[13]. Moreover, much effort is put into improving electrostrictive performance by inducing relaxor phase in PVDF through electron irradiation[17,18] or other surface modification methods[19]. Nevertheless, these methods only affect the region near the film's surface. The corresponding improvement in electrostriction is still unsatisfactory, usually less than one-tenth of the value accomplished by the cross-linking method.

In fact, the dilemma for further promoting the electrostriction and the overall EM performance of PVDF lies in its semicrystalline structure, i.e., the different responses of crystalline and amorphous phases under the electric field[20,21]. It is believed that the limited improvement of electrostriction on the crystal phase should stem from the insufficient charge transport capacity of the amorphous phase[22]. Due to the Maxwell–Wagner effect, the charge will segregate

[1]State Key Laboratory of Advanced Design and Manufacturing Technology for Vehicle, College of Mechanical and Vehicle Engineering, Hunan University, Changsha, China. [2]Key Laboratory of Advanced Design and Simulation Technology for Special Equipment, Ministry of Education, College of Mechanical and Vehicle Engineering, Hunan University, Changsha, China. [3]Institute of Micro/Nano Materials and Devices, Ningbo University of Technology, Ningbo, China. ✉e-mail: nzhang@hnu.edu.cn; jiangc@hnu.edu.cn

on the crystal grain boundary with the deflection of dipoles during electro-deformation. These segregated charges are hard to dissipate through the surrounding amorphous phase. Hence, the elevated inner potential inside the crystal grain impedes further deflection of the dipoles and thus suppresses the electrostrictive and dielectric performance of the crystal phase. In other words, enhancing charge transport capacity at the amorphous region may become a promising method to exploit the crystal phase's deformation potential.

Furthermore, although the effect of electron irradiation on electrostriction is limited, the vastly improved dielectric property implies that the deformation potential of the crystal phase can be exploited by charge injection[23,24]. It has been reported that an electret structure with high charge density is beneficial to the charge transport capacity of the amorphous phase[25,26]. However, traditional electrets are generally introduced in a top-down approach[27,28], i.e., the charge is injected through post-processing after the formation of the film. Therefore, the affected area is usually limited to the film's surface[29], and the advantage of the improved charge transport capacity cannot extend throughout the film, especially under a low electric field[30].

To break through this limitation, we expect to create an electret structure inside the material to obtain a high charge density at the interior. This mission is accomplished by introducing the charge in a bottom-up approach[31], i.e., during the printing of PVDF films. The new electret type is called the Bottom-Up Electret (BUE) and should reduce the hindrance of crystal deflection throughout the film. Consequently, the BUE-PVDF fabricated with charge printing is expected to enhance the electrostrictive performance and pave a novel way to improve the overall EM performances of the advanced polymer materials.

## Results
### Design and fabrication of the BUE-PVDF films

Figure 1A schematically shows the formation process of BUE-PVDF film with the electrohydrodynamic (EHD) printing technology. The deposited droplets on the substrate form Z-shaped tracks, and the layer-by-layer accumulation of the tracks produces a mesh-like topography structure. At the same time, the charge carried in ink will accumulate within this stacking of the membrane material. For most printed layers at the interior of BUE-PVDF, uniformly distributed high-density positive charges (Interior charge) are expected to be stored inside the material. Meanwhile, negative charges (Surface charge) with proportional density should be absorbed from the air to the outmost layers on both surfaces to balance the inner electric potential. Given that the printed charge tends to dissipate outside the film during printing[32], the naturally formed pore structures (inset of Fig. 1A) could act as traps to store the printed charge in this study. By printing and preserving high-density charge throughout the whole material, the deliberately designed BUE structure could fully exploit the deformation potential of crystal grains at the interior of PVDF films.

The X-ray diffraction (XRD) results of BUE-PVDF with different printing voltages are shown in Fig. 1B. A considerable amount of crystal phase is retained after forming the BUE structure, which meets the fundamental prerequisite to determine most EM properties of PVDF films. The scanning electron microscopy (SEM) surface topographies (Fig. 1C–F) display the size and distribution of pore structure in BUE-PVDF samples. The surface of BUE-PVDF generally exhibits a mesh-like pattern that stems from the Z-shaped printing tracks (Fig. 1C, yellow and white dashed lines). There are two different regions with distinct pore structures depending on whether the region is located on the ink tracks (Fig. 1D). The size of pores at the gap area can be over 10 μm

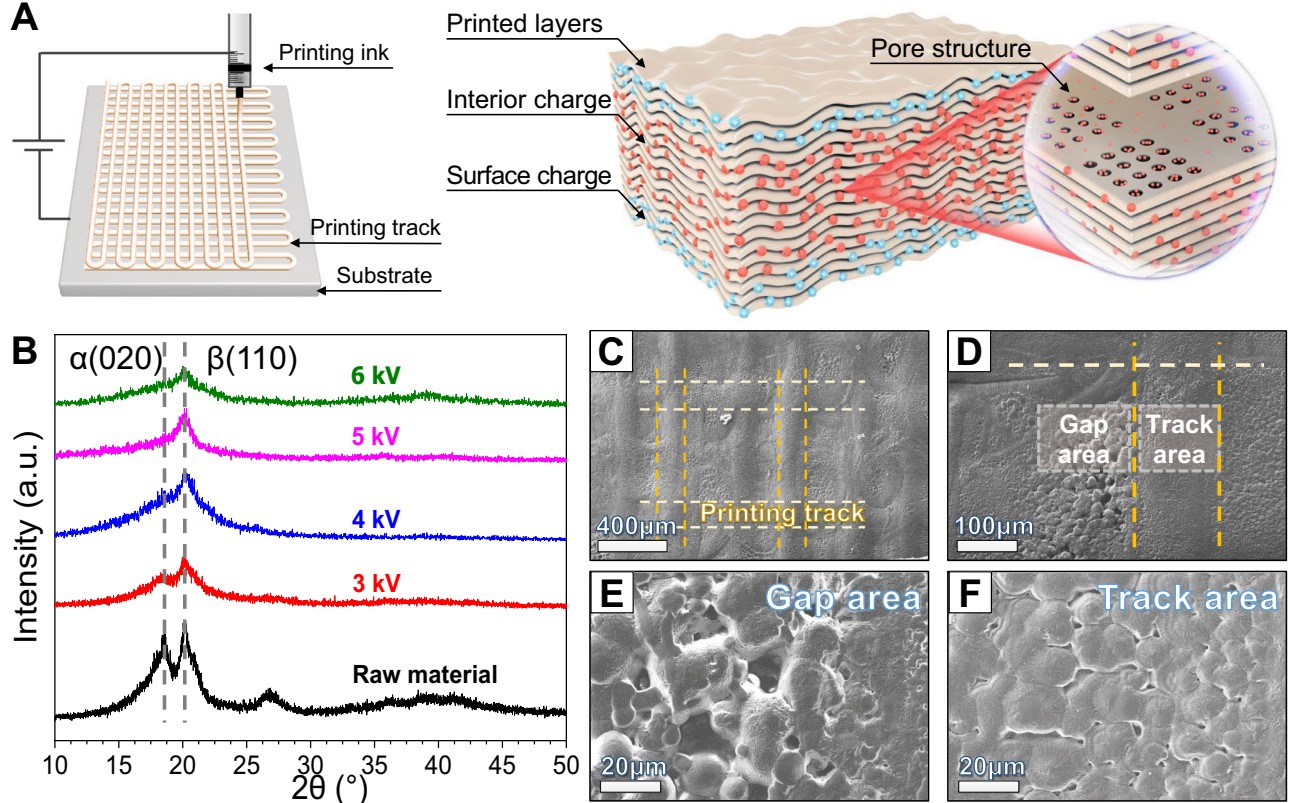

**Fig. 1 | Design and fabrication of the BUE-PVDF films. A** The basic process of EHD printing with the Z-shape track on the substrate and the scheme of the BUE structure with evenly distributed charge and porosities. **B** The XRD pattern of 14 wt % PVDF in NMP under different voltages from 3 kV to 6 kV. **C** The SEM image of the printed PVDF samples with its ink track pattern. **D** Zoomed-in views of the gap and track area around the ink track. The SEM images of the printed PVDF samples with even higher magnification show the pore distribution within the gap area (**E**) and track area (**F**), respectively.

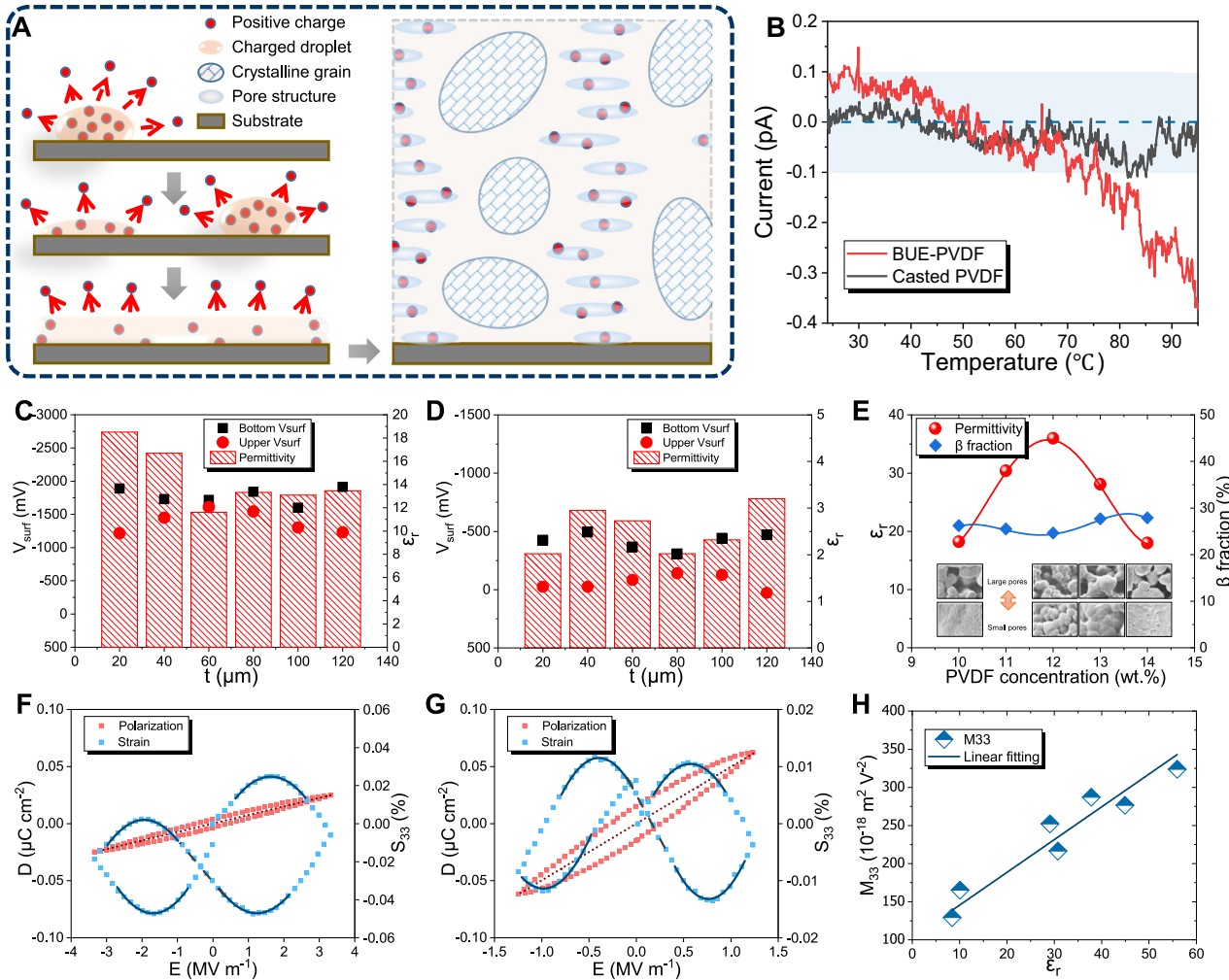

**Fig. 2 | The correlation among charge density, dielectric, and electrostrictive properties. A** Scheme of the storing mechanism of printed charge during the printing process. **B** The TSC curve for the printed (red) and casted (black) PVDF film, respectively. **C** The KPFM potential $V_{surf}$, tested on both sides of the printed PVDF films with varied film thickness $t$, along with the red column showing each sample's relative permittivity $\varepsilon_r$. **D** The KPFM potential test results for samples with lower relative permittivity. **E** The relative permittivity and $\beta$ fraction of the PVDF films with varied concentrations (from 10 to 14 wt%) in NMP solvent. **F** The polarization hysteresis analysis of the printed PVDF sample with $\varepsilon_r = 8.8$. **G** The polarization hysteresis analysis of printed PVDF sample with $\varepsilon_r = 56$. **H** The relationship between electrostrictive coefficient $M_{33}$ and the relative permittivity $\varepsilon_r$ in the EHD printed PVDF films.

(Fig. 1E), which is much larger than the size at the track area (Fig. 1F, ~0.5–3 μm).

## The correlation among charge density, dielectric, and electrostrictive properties

A schematic diagram in Fig. 2A elaborates the mechanism for trapping the printed charge by the pore structure. During solidification, most of the positive charge in the charged droplet will dissipate at the free surface. Meanwhile, part of the charge is trapped by defects within the film, especially on the surface of the pores. With the accumulation of printing layers, these trapped charges are not able to escape from these pores surrounded by the insulated amorphous phase. Therefore, the charge density should increase with the density of the pores, which is verified by the different results of open-circuit thermally stimulated current (TSC) tests between the BUE-PVDF and solvent-casted sample (Fig. 2B). The results show that BUE-PVDF continuously releases a large amount of interior charge during heating from 30 °C to 90 °C. This interior charge is more easily released at a relatively higher temperature (>70 °C), with the releasing current value increasing from 0.1 pA to 0.3 pA. In contrast, the casted sample, which carries a current consistently less

than 0.1 pA, hardly releases interior charge at the same temperature range.

Since pores inside the PVDF can effectively store the printed charge, controlling pore structure during the printing process could tailor the charge density and thus improve the EM properties in PVDF films. Figure 2C, D compare the Kelvin probe force microscopy (KPFM) and dielectric test (by impedance method) results of two PVDF groups with different charge densities. These results demonstrate that the upper and lower surface potential and dielectric measurements are almost unrelated to the film thickness. More importantly, the average surface potential increases with the relative permittivity. For instance, the average surface potential of the sample from Fig. 2C (1.59 V) is 6.4 times that of the sample from Fig. 2D film (0.25 V), and its relative permittivity (14.4) is also 5.8 times that of the latter (2.5). This relation between the surface potential and relative permittivity successfully validates the significant effect of charge density on the dielectric property. Notably, the surface potential of BUE-PVDF samples is below 5 V, much smaller than traditional electrets' surface potential (greater than 500 V). This remarkable discrepancy in surface potential implies that the proposed new electret structure is much more stable and adaptable in varied practical applications than traditional electrets.

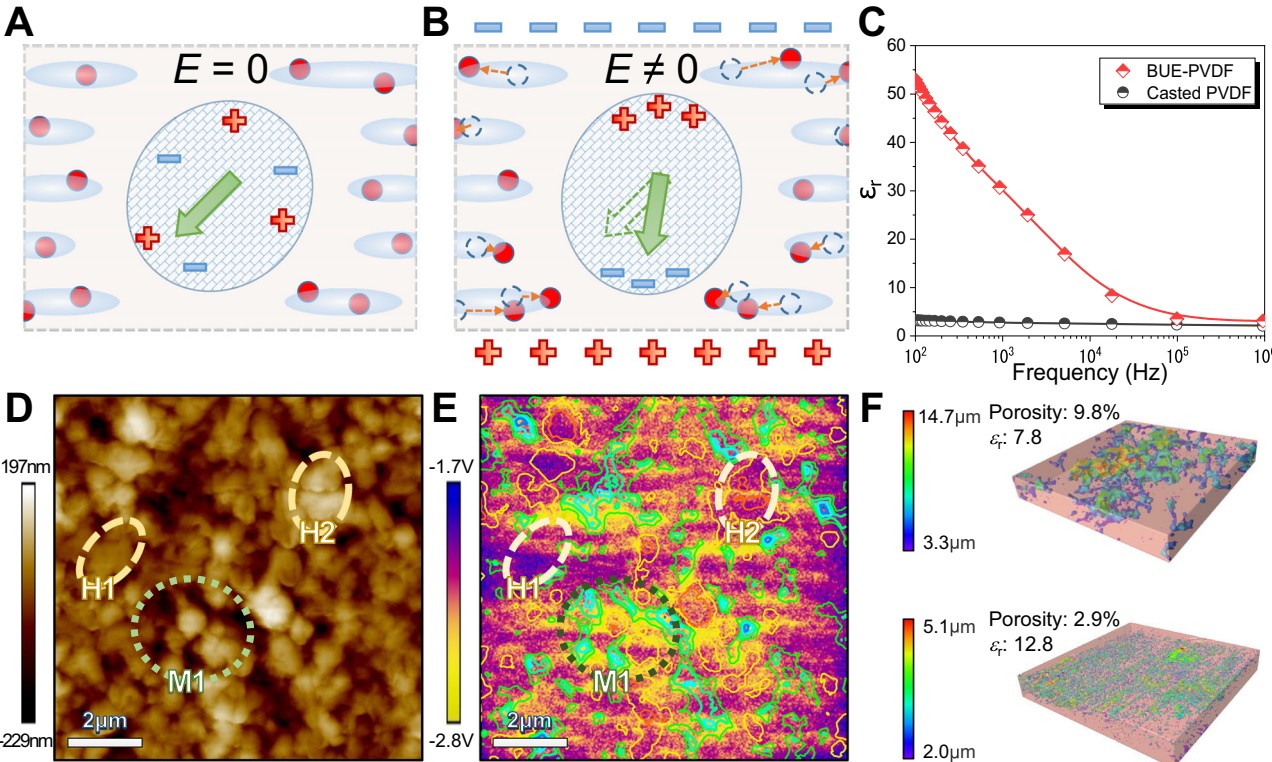

**Fig. 3 | The origination of ultra-high electrostriction in the BUE-PVDF film.**
**A, B** the schemes of the electro-lubricating effect in BUE-PVDF showing crystal deflection behavior with or without the applied field. **C** Measured frequency response of dielectric property $\varepsilon_r$ for BUE-PVDF (red curve) and casted PVDF (black curve). **D** The topography (surface height) image on the surface of BUE-PVDF film with pore structure. **E** The comparison diagram between the contour plot of surface height and potential image to demonstrate the stored charge around pores. **F** The 3D microcomputed tomography reconstruction of the PVDF film structure with varied porosity and permittivity.

Figure 2E summarizes the change of relative permittivity in BUE-PVDF films with varying PVDF concentrations to further study the effect of pore structure on the dielectric property. The sample with a moderate PVDF concentration (12 wt%) exhibits the highest pore density (inset of Fig. 2E) and relative permittivity (36). Further increase or decrease of PVDF concentration reduces its dielectric property (see Note 9 of Supplementary Information for detailed information). Regarding the pore size distribution, all four samples have plenty of large pores, and their relative permittivity is above 15. Two samples (12 wt% and 13 wt% concentration) with plenty of small pores have a relatively higher permittivity value of 36 and 28, which is 1.5–2 times the value of the other two samples with almost no small pores. Therefore, pores with smaller sizes are more effective in improving the dielectric property of BUE-PVDF. We should highlight that the $\beta$ phase fraction is almost constant with varying concentration, indicating that the performance improvement in BUE-PVDF does not originate from the traditional $\beta$ phase enhancement.

The ferroelectric analysis is carried out on BUE-PVDF films with varied permittivity to verify the influence of dielectric property on electrostriction (Fig. 2F, G). The relative permittivity and the inverse piezoelectric coefficient ($d_{33}$) are evaluated by measuring the slope of the displacement-electric field (D-E) curve (red dotted line) and the slope of the strain-electric field (S-E) curve near the origin (gray dash line), respectively. The electrostrictive coefficients ($M_{33}$) are obtained by fitting the S-E curve's quadratic term (solid blue line). With the increasing relative permittivity from 8.8 (Fig. 2F) to 56 (Fig. 2G), the optimized values of $M_{33}$ increases accordingly from $129 \times 10^{-18}$ m$^2$ V$^{-2}$ to $324 \times 10^{-18}$ m$^2$ V$^{-2}$ (Detailed discussion about the optimized permittivity and electrostrictive coefficient is given in Supplementary Information). Also, a giant $d_{33}$ value of 298 pm V$^{-1}$ is observed from the sample with optimized electrostriction in Fig. 2G.

The dependence of electrostrictive coefficient on dielectric property is further examined and confirmed in Fig. 2H. With the relative permittivity in BUE-PVDF films improving from 8.8 to 55.9, the corresponding value of their electrostrictive coefficient linearly increases by ~2.5 times. This linear relation between electrostrictive coefficient and dielectric property is believed to stem from the correlation between EM properties and the crystal phase, which is only reported in ceramic electrostrictive materials[12]. However, the insulated amorphous phases prevent charge transportation and the deflection of internal crystal grains within the polymer materials[29]. The overall electrostriction in polymer materials cannot markedly improve with the dielectric property commonly measured on the surface. Hence, the linear dependence of electrostrictive coefficient in Fig. 2H implies that the BUE structure creatively enhances the capacity of charge transportation around the crystal grains, thus, the EM properties throughout the PVDF film.

## The origination of ultra-high electrostriction in the BUE-PVDF film

Figure 3A, B illuminate the effect of the printed charge upon the crystal phase in BUE-PVDF. The pores and printed charges are randomly distributed at the surrounding amorphous matrix for a single crystal grain in BUE-PVDF (Fig. 3A). The dipoles inside the crystal are deflected with the increasing applied electric field (Fig. 3B), resulting in segregated charges on the grain boundary and an inner potential opposite to the applied electric field. Due to the Maxwell–Wagner effect, this inner potential prevents the crystal from further deflection and suppresses the EM performances. However, the inner potential also affects the printed charges around the crystal. By traveling across the pore surface, these localized space charges counteract the effect of the segregated crystal charges, which is equivalent to the dissipation of crystal

charges by the surrounding amorphous phase. Hence, the printed charges reduce the hindrance of crystal deflection and thus exploit the deformation potential of the crystal phase in BUE-PVDF. This mechanism is therefore named as the electro-lubricating effect.

Given the difficulties of in-situ observation, we can indirectly verify the origination of ultra-high electrostriction in BUE-PVDF from the electro-lubricating effect through the following micro-scale experimental phenomenon. Firstly, the enhanced relative permittivity should stem from the rising density of the printed charge (Fig. 2C, D). Meanwhile, a large number of the printed charges should be stored around the pore surface so that the charge density increases with the density of the surface pores (Fig. 2E). Besides, the stored charges and pore structure should coexist on both the surface and interior of BUE-PVDF to enhance the permittivity throughout the film, guaranteeing the ultra-high electrostrictive coefficient and its linear dependence on the permittivity (Fig. 2H).

The origin of the improved dielectric property in BUE-PVDF is investigated by examining permittivity with varying electric field frequencies through an impedance analyzer in Fig. 3C. Since each mechanism contributing to the overall permittivity has a characteristic cut-off frequency, the low-frequency mechanisms drop out in turn with the frequency increase[33]. Compared to its casted counterpart, the extra contribution to permittivity in BUE-PVDF nearly ceases to exist when the frequency is above $10^6$ Hz. This frequency range coincides with the traveling charge carrier mechanism produced by space charge or interfacial polarization[33], suggesting that the improved dielectric property of BUE-PVDF mainly originate from the printed charge.

To verify the storing of the printed charges on pore surfaces, the surface topography and potential distribution in BUE-PVDF samples are examined by the KPFM test (Fig. 3D, E). As shown in Fig. 3D, there are multiple dark areas around the light-colored area on the sample surface. These dark pores are similar to those small pores shown in Fig. 1F. To examine the storage of printed charges, the contour plot converted from the topography plot in Fig. 3D is superimposed on the corresponding surface potential distribution plot in Fig. 3E. The middle-density region (e.g., area enclosed by green dotted curve M1 in Fig. 3D) with more pores (green/blue contours in Fig. 3E) often accompanied with stored charges (yellow color area in Fig. 3E with an absolute potential value higher than 2.2 V). In contrast, the high-density region (e.g., areas enclosed by yellow dash curve H1 and H2 in Fig. 3D) away from the pore structure have relatively lower charge density (dark color area in Fig. 3E). This result suggests that the printed charges tend to store around the pores. Meanwhile, more KPFM results in Supplementary Information further verify such storing effect.

Microcomputed Tomography (Micro-CT) scans are carried out for samples with different pore sizes to investigate the distribution of pore structure at the interior of BUE-PVDF. Figure 3F displays the representative cuboid ($200 \times 200 \times 28 \mu m$) reconstructed from the internal region of a BUE-PVDF film with about 80 μm in thickness. The cross-sections of the reconstructed region along the thickness direction in Fig. 3F are given in Supplementary Movie 1 to further exhibit the pore distribution. Since most pores within the PVDF film are naturally introduced during the layer-by-layer printing process, the pore distribution at the interior of the film (Fig. 3F) should be comparable to that on the surface (Fig. 2E). Given the correlation among the pore structure, the charge density, and the dielectric property on the surface of BUE-PVDF displayed in Fig. 3C, E, the existence of numerous pores in Fig. 3F suggest strongly that the electro-lubricating effect improves the dielectric performance throughout the whole film. Note that the sample containing small pores with lower porosity (2.9%) achieved higher relative permittivity (12.8) in Fig. 3F. This is probably due to the higher electro-lubricating efficiency on crystal grains by smaller pores under the same porosity or even the same total surface area (Detailed explanation is given in Note 12 of Supplementary Information). Therefore, forming a more dispersed pore structure to tailor the charge distribution within the printed films can become a creative approach to improve the EM performance in PVDF.

## The comparison of the BUE-PVDF with other electroactive materials

The proposed enhancing mechanism for the electrostriction differs fundamentally from all previously reported methods in PVDF. Figure 4A exhibits the dielectric and electrostrictive properties of various PVDF and ceramic materials. For the PVDF-based polymer improved by the irradiation (purple diamond) method, the relative permittivity was distributed from 8 to 56.5. However, the lower internal dielectric property limits its overall electrostrictive performance (less

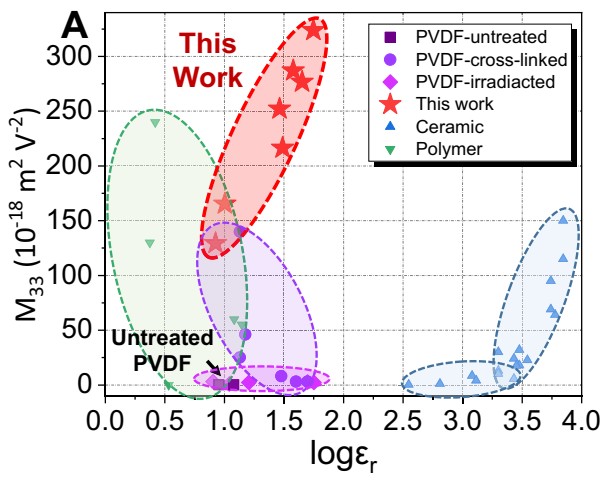
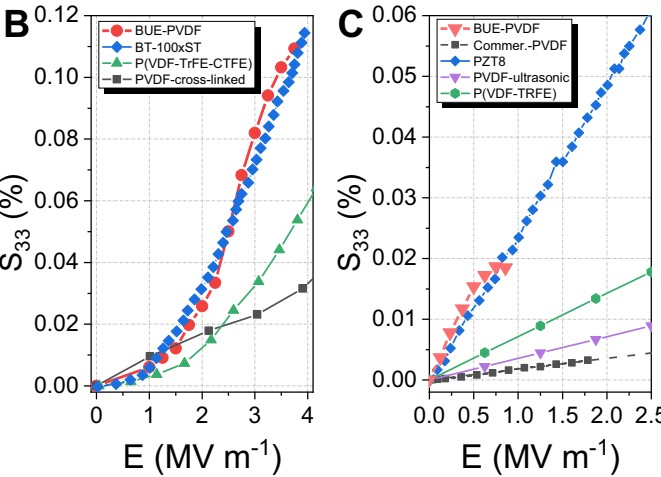

**Fig. 4 | The comparison of the BUE-PVDF with other electroactive materials concerning their electrostrictive coefficient ($M_{33}$), relative permittivity ($\varepsilon_r$), and induced strain ($S_{33}$). A** The plot of the relative permittivity against electrostrictive coefficient for the present PVDF with optimized BUE structure. Other relative progresses in PVDF-based polymers[14,17,18,23,24,35,37,45,63–70], other polymers[37], and ceramics[38–43,71–73] are also incorporated for comparison purposes, confirming the excellent dielectric and electrostrictive properties of the present BUE-PVDF. **B** The comparison of the electrostrictive strain of the present BUE-PVDF with the electrostrictive ceramic BT-100xST under a low electric field $E$[46], the cross-linked PVDF by plasticizer[48], the cross-linked P(VDF-TrFE-CTFE) terpolymer[16], and **C** The comparison of the piezoelectric strain of the present BUE-PVDF with commercial β-phase PVDF, piezoelectric ceramic PZT-8[74], and advanced PVDF-based piezoelectric polymer by multiple methods[61,75].

than $5 \times 10^{-18} \, \mathrm{m^2 \, V^{-2}}$). The PVDF-based polymer enhanced merely by cross-linking (purple circle) promotes the electrostrictive performance to a much higher level of $140 \times 10^{-18} \, \mathrm{m^2 \, V^{-2}}$ due to its influence throughout the film[14]. However, the relative permittivity does not exceed 15 when the electrostrictive performance is above $25 \times 10^{-18} \, \mathrm{m^2 \, V^{-2}}$ [16,34,35]. Thus, improved electrostrictive performance is usually achieved at the cost of crystallinity, which will lead to lower dielectric and other EM properties. This trade-off between electrostrictive and dielectric properties has also been found in several electrostrictive polymers (green triangle)[36,37].

In contrast, BUE-PVDF enhanced by the unique electret structure simultaneously improves electrostrictive and dielectric properties. Previously, this positive correlation was commonly applied in ceramic materials (blue triangle) to promote electrostriction by optimizing dielectric property[12,38–43]. In the present work, the electrostrictive coefficient $M_{33}$ in BUE-PVDF is upgraded to a maximum value of $324 \times 10^{-18} \, \mathrm{m^2 \, V^{-2}}$ with the relative permittivity optimized to 56. Such an ultra-high value of $M_{33}$ is $10^4$ times the original PVDF (purple square)[44], more than 100 times the irradiated PVDF[17], and more than two times the highest value in all PVDF-based materials[45].

In addition to the ultra-high electrostrictive coefficient, the high strain response behavior under a low electric field is another significant advantage for BUE-PVDF. Figure 4B compares the strain behavior of BUE-PVDF and electrostrictive ceramic materials under an electric field below $4 \, \mathrm{MV \, m^{-1}}$. The electrostrictive strain of BUE-PVDF at this field range is almost equivalent to BT-100xST ceramic material[46]. The operative electric field required to achieve a 0.11% strain in BUE-PVDF is $4 \, \mathrm{MV \, m^{-1}}$, while such value in ordinary PVDF is higher than $25 \, \mathrm{MV \, m^{-1}}$ [47]. High electrostriction in PVDF at this field range is previously achieved by cross-linking either using a high content of the plasticizer tricresyl phosphate (TCP) (the black squares in Fig. 4B, $M_{33} = 16 \times 10^{-18} \, \mathrm{m^2 \, V^{-2}}$)[48] or forming the ter-polymer/multi-polymer (the green triangles in Fig. 4B, $M_{33} = 44 \times 10^{-18} \, \mathrm{m^2 \, V^{-2}}$)[16]. Although not being fully optimized, the BUE-PVDF in Fig. 4B still has an obvious advantage over these two types of materials as shown (the red circles, $M_{33} = 98 \times 10^{-18} \, \mathrm{m^2 \, V^{-2}}$). This remarkable reduction of the operative field in BUE-PVDF would significantly broaden the promising application of PVDF-based polymers.

The high electrostriction under a lower electric field could further lead to ultra-high ferroelectric performance in BUE-PVDF. Figure 4C compares the piezoelectric strain of BUE-PVDF with optimized piezoelectric PVDF and typical piezoelectric ceramic PZT-8. BUE-PVDF exhibits an ultra-high inverse piezoelectric coefficient ($d_{33}$) of $298 \, \mathrm{pm \, V^{-1}}$ (Fig. 2G). This value is about 10 times the traditional limit for the $d_{33}$ value in pristine PVDF[49] and is even comparable to PZT-8 (Fig. 4C). Note that previous efforts to improve the piezoelectric performance in PVDF are all based on enhanced polarization and $\beta$ crystal phase[11], mainly including the annealing method to enhance the $\beta$ phase fraction[50,51], the poling method to induce the $\beta$ phase and polarization altogether[52–54], the stretching method to facilitate the phase transformation from $\alpha$ to $\beta$ phase[55,56], the electrospinning method to prepare piezoelectric nanofibers in in-situ stretching and poling at the same time[57,58], and chemical modification to increase the TTTT conformation chain (which will lead to higher $\beta$ fraction in its crystal phase) for PVDF[59,60]. In recent years, the best $d_{33}$ is obtained around $63.5 \, \mathrm{pm \, V^{-1}}$ by distinctly tuning the composition of TrFE monomers to induce the morphotropic phase boundary (MPB)[61] (Fig. 4C). Although Chen et al. even reported a PVDF-based material with an ultra-high $d_{33}$ of up to $1050 \, \mathrm{pm \, V^{-1}}$ [62], the bias electric field used in this work is as high as $30\text{–}40 \, \mathrm{MV \, m^{-1}}$. In contrast, the extraordinary EM performance in BUE-PVDF originates from the electro-lubricating effect, which markedly lowers the deflection resistance of dipoles. Therefore, it does not require post-polarization or bias electric field to achieve ultra-high piezoelectric performance.

## Discussion
In summary, this study innovatively designed a BUE structure by controlling the pore structure to realize the high density of printed charge at both the surface and interior of the PVDF films. This high density of printed charge activates the electro-lubricating effect, i.e., facilitating the electrostrictive deformation of the crystal phase by the nearby space charge. With the help of the BUE structure, a great promotion of dielectric, electrostrictive, and piezoelectric properties is simultaneously achieved in PVDF. Furthermore, the electric field required for its noticeable strain in BUE-PVDF is greatly reduced to less than $4 \, \mathrm{MV \, m^{-1}}$, which significantly expands the application field of PVDF as component in electronic devices. Moreover, the essential function of BUE is to fully activate the intrinsic EM potential within the material itself, especially for the interior part of the film. Hence, an ultra-high overall strain (>1%) has not been obtained in the pristine PVDF with a BUE structure under a higher electric field. Suppose we introduce BUE structure in electroactive material with higher deformation potential (e.g., P(VDF-TrFE) or multi-polymers of PVDF). It is reasonable to anticipate more incredible improvement of EM properties in this material class. Therefore, the charge printing technology and the electro-lubricating effect are expected to realize much better EM performances for PVDF and bring a revolutionary change in the fabrication and promotion of advanced polymer materials.

## Methods
### Preparation of PVDF ink
In a typical procedure, 0.8–1.4 g PVDF powder (with $M_w$ of $534{,}000 \, \mathrm{g \, mol^{-1}}$ or $275{,}000 \, \mathrm{g \, mol^{-1}}$, respectively) was added to 10 g of N-methyl-2-pyrrolidone (NMP ≥ 99%) or N, N-Dimethylformamide (DMF ≥ 99%) and then ultrasonically stirred at room temperature. After complete dissolution, 10 g of the above solution and 0–1 g of Acetone (≥99.5%) were mixed, followed by magnetic stirring for 1–4 h to ensure the complete mixing of all components in the solution. At this stage, the ink was kept under yellow light conditions to prevent unnecessary exposure. PVDF powders were purchased from Arkema Co., Ltd. Acetone and NMP solvent were provided by China National Pharmaceutical Group. All chemicals were directly used as received. The detailed procedure for the 3D printable PVDF ink preparation was illustrated in Supplementary Information.

### EHD printing procedure
EHD printing of the PVDF ink was conducted based on a custom-designed 3D printer system (A3S, Shenzhen Aurora Technology Co. Ltd) with various sizes of nozzles (23#–30#). Printing paths were generated by CAD drawings (SolidWorks, Dassault Systemes) and converted into G-code by a commercial software package (A3 Cura, Aurora Tech) and custom Python scripts to command the x-y-z motion of the printer head. The EHD printing was conducted at 3–8 kV with a high-voltage power supply (Dongwen High Voltage Power Supply (Tianjin) Co., Ltd) connected to the needle. To achieve constrained drying of the printed PVDF film in the thickness direction, the samples were placed on a glass substrate and dry-annealed to ensure all solvent was removed from the sample. Finally, the obtained films, with a thickness of ~20–120 μm each, were peeled off the substrate.

### Structural characterization
XRD θ-2θ scans at room temperature were acquired using a Bruker D8 Advance X-ray Diffractometer in the Bragg-Brentano geometry with a source emitting Cu K$_\alpha$ radiation (wavelengths $\lambda_{K\alpha} = 1.54$ Å). The surface morphology of PVDF samples was analyzed through Field Emission Scanning Electron Microscope (FE-SEM). An FE-SEM (Sigmal 300, Carl Zeiss, Oberkochen, Germany) operating with an accelerating voltage of 2 kV was used to assess the morphology of the PVDF films. A Quorum Technologies Q150R-ES sputter coater (Laughton, East Sussex, UK) was used to metalize the PVDF films before SEM imaging with

20 nm of Au, to prevent charging. Internal structure characterization was conducted and reconstructed through microcomputed tomography (Micro-CT) equipment (Zeiss515 Versa, Germany).

## Electro-mechanical (EM) and dielectric measurements

Gold electrodes with a typical thickness of 60 nm were sputtered (using a Q150R-ES sputter coater) on both sides of the polymer films for electrical measurements. Dielectric spectra were acquired over a broad temperature range using the dielectric Wayne Kerr 6500B Series Impedance Analyzer. The permittivity $\varepsilon$ and loss (tan$\delta$) were measured as a function of frequency $f$ (100–1,000,000 Hz) at a voltage of 0.5 V. The dielectric property (relative permittivity) was also measured from the slope of the $D$-$E$ curve in ferroelectric analysis. Note that unless otherwise specified, all the permittivity values in this work are measured using the ferroelectric method. To measure bipolar $D$-$E$ field loops, a Sawyer-Tower circuit was used, where the PVDF films with electrodes were subjected to a triangular bipolar wave. The electrostrictive behavior of PVDF film was evaluated by measuring the induced strain ($S$) by a photonic displacement sensor (MTI-2100 Fotonic sensor, sensitivity 0.0058 mm mV$^{-1}$) at 10 Hz in a low electric field ($E \leq 100$ MV m$^{-1}$). The electric field ($E$) was applied in the thickness direction using a high-voltage amplifier (Trek model 610E). The electrode displacement was determined by the photonic sensor and lock-in amplifier, which were controlled by the LabVIEW computer program. The electrostrictive coefficient ($M_{33}$) was calculated from strain and electric field strength. All measurements were made at ambient room temperature (20–25 °C).

## Charge density measurement

TSC measurements were performed on PVDF film samples to investigate the amount of charge stored in the film and how it dissipates with increasing temperature. During the TSC experiment, we place the charged sample in a temperature-controlled test chamber. The single-surface electroded sample was mounted between two copper plates, open-circuited, and held at 23 °C for 1 h to release the surface charge, and then linearly increased from 23 °C to 140 °C at a rate of 3 °C min$^{-1}$. The exact current signal was then obtained with the help of a current amplifier (Keithley 428). At last, these current values were collected by the Keithley-6517B data acquisition recorder to generate the overall thermal current spectrum.

## Surface potential measurement

KPFM surface potential measurements of film were performed with a Dimension Icon AFM (Bruker Corporation, Germany) using (tap150E-G) with a resonance frequency of 150 kHz and a spring constant of 5 N m$^{-1}$. In this work, we applied a two-pass amplitude-modulated KPFM (AM-KPFM) to measure the surface potential of the sample (test environment: H$_2$O < 1 ppm, O$_2$ < 1 ppm). During the first pass scan, we applied the standard alternating current (AC) mode imaging (with a typical tip oscillation amplitude of 20 nm) to obtain the topography and phase signal of the film sample. In the second pass scan, the tip of the needle was raised to a certain height (typically 80 nm), and this scan was made according to the topographic lines obtained from the first scan. Data about the surface potential can be obtained by detecting the direct current voltage applied to the tip of the needle, which nullifies the tip's interaction with the sample.

## Mechanical properties characterization

The mechanical properties of the 3D printed specimen (20 mm × 20 mm × 40 μm by design) were tested using a mechanical testing system (SHIMADZU, Model SEM-SERVO PULSER). The specimen was uniaxial stretched along the X-direction at a rate of 0.5 mm min$^{-1}$ until sample fracture was detected in the stress-strain plot. Young's modulus was obtained by calculating the slope of the initial linear region of the stress-strain curves.

## Data availability

The data supporting the findings of this study are available in the paper and the Supplementary Information and can be obtained upon request from the corresponding author nzhang@hnu.edu.cn. Source data are provided with this paper.

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

## Acknowledgements

C.J. acknowledges the financial support received from the National Natural Science Foundation of China (52235005); the XPLORER PRIZE and New Cornerstone Science Foundation. S.H. acknowledges the financial support received from the National Natural Science Foundation of China (51901075). We thank Dr. Weiyou Yang for his insightful discussions on this paper and Dr. Lejia Wang for the AFM-PFM test system at the Institute of Micro/Nano Materials and Devices, Ningbo University of Technology. We thank Dr. Gangjin Chen for the TSD instrument at Hangzhou Dianzi University. We thank Dr. Qingling Chen for the assist with micro-CT testing at Analytical Instrumentation Center of Hunan University. We thank Dr. Yi Yin for the assist with space charge measurement and insightful discussion at Shanghai Jiao Tong University. We thank Dr. Yuanying Yu for the assist with *P-E* curve testing and analysis at Wuhan University of Technology. We also thank Dr. Zhi Fang for the help in drawing work at Peking University.

## Author contributions

Supervision, C.J.; Conceptualization, N.Z., S.H., and C.J.; Methodology, N.Z.; Investigation, N.Z., X.D., Z.L., and Q.Q.; Visualization, S.H., X.D., and Q.Q.; Writing—original draft, N.Z., S.H., and W.L.; Writing—review and editing, N.Z., S.H., W.L., and C.J.

## Competing interests

The authors declare no competing interests.
