## [Transparent Peer Review file · Nature Communications]

Ultra-high electrostriction and ferroelectricity in poly (vinylidene fluoride) by 'printing of charge' throughout the film

Corresponding Author: Professor Chao Jiang

Version 0:

Reviewer comments:

Reviewer #1

(Remarks to the Author)

Electroactive polymers with high ferroelectricity and large electrostriction are highly desirable for flexible devices. The authors, however, have presented a truly unique strategy in this work. They have explored high-performance PVDF-based materials not through the conventional optimization of the crystal phase or the adjustment of the carbon chain structure, but through the synergism of the crystal and amorphous phases. This novel approach, coupled with 3D printing to introduce a new structure, has fully exploited the benefits of crystal grains throughout the PVDF film. The measured electrostrictive coefficient and piezoelectric constant are notably higher than the values reported in the literature, indicating a significant advancement in this field. This work also provides fresh insight into the correlation between dielectric, electrostrictive, and ferroelectric properties in electroactive polymers. Therefore, I highly recommend it for publication in Nature Communications once the following issues are addressed.

- 1) The authors mention ultra-high ferroelectricity in the title and abstract. Please briefly introduce the techniques and principles used to enhance ferroelectricity in PVDF materials.
- 2) This article points out that increased pore density improves the dielectric properties of materials in Fig. 2E. Could it be possible that such improvement is due to the increased conductivity caused by the high porosity?
- 3) In Fig. 1, the authors classify the porosity structures as gap and track area but never mention this classification again in the following characterization results (e.g., Fig. 2E). An explanation is suggested.
- 4) Since Fig. 2F and 2G demonstrate a high inverse piezoelectric effect, please explain why the corresponding hysteresis curves are not similar to those of the typical ferroelectric material.
- 5) In Fig. 4B, we find a typical parabolic (electrical field vs. strain) curve from BUE-PVDF, while the curve changes linearly in Fig. 4C. What factors result in such a difference?
- 6) In Fig. 4C, please include a more detailed discussion about other PVDF-based materials to compare the strategies used to enhance ferroelectricity.

Reviewer #2

(Remarks to the Author)

Comments

This manuscript describes the fabrication of 3D electret structure consisting of PVDF via electro hydrodynamic printing to attain maximum electrostriction effects. Various samples with different ink concentrations, printing times, voltages and thicknesses were experimented. Through surface printing of charges utilizing the bottom-up technique, the authors could obtain a maximum electrostrictive coefficient value of $324 \times 10^{-18} \text{ m}^2/\text{V}^2$ which is 104 times greater, and attained a piezoelectric constant d_{33} of 298 pm/V which is close to 10 times than the commercial PVDF films, making them suitable for use in smart devices and systems.

1. The authors should cite the previously reported electrostrictive coefficients observed for PVDF-based samples obtained through surface modification and electron irradiation for effective comparison with the printed sample.
2. On which substrate was the EHD printing carried out before placing it onto the glass substrates?
3. Though ferroelectric studies were carried out to better reflect dielectric properties, whether the authors tried to compare the

results by measuring the same through impedance analysis?

4. Reason for specifically choosing 534000 g/mol and 275000 g/mol molecular weight of PVDF is not substantiated. How the ink concentrations were fixed for all samples?

5. Can the authors contrast the effect of PVDF blending in sample 5 with other samples (1-4)?

6. In general porous structure leads to charge loss, how efficient the printed electrode?

7. Any changes in the crystallinity noticed from KPFM studies between BUE-PVDF and Solvent-casted PVDF which may have a direct effect on the surface activity?

8. According to Fig 2c, as relative permittivity is greater for 12 wt % of PVDF printed sample, charge density is greater for that sample. Whether the surface potential measurements were done for the corresponding sample to observe its pore density?

9. From Fig. 3F, it is seen that medium sized pores are only agglomerated on the surface and near surface, whereas low density pores (blue) are evenly distributed in the interior of the samples. How do the authors correlate the charge density and pore density based on this?

10. Graphs pertaining to strain behaviour as a function of electric field, electrostrictive coefficients and absolute β fraction should be provided.

11. What was the optimized PVDF wt % for relative permittivity value of 56?

12. What was the value of electrostrictive coefficient for solvent casted PVDF film?

Version 1:

Reviewer comments:

Reviewer #1

(Remarks to the Author)

Publish as is

Reviewer #2

(Remarks to the Author)

Comments

Though authors have considerably answered all the queries, there are few other comments which warrants further clarification.

1. The authors should better clarify comment no. 5, as the effect of blending two different molecular weight PVDF is not effectively contrasted with other samples. It would be better, if authors can compare in terms of performance improvement with other samples.

2. The optimized PVDF wt % for relative permittivity value of 56 in comment no.10 is also not clearly substantiated.

Version 2:

Reviewer comments:

Reviewer #2

(Remarks to the Author)

The manuscript has been revised well. It can now be considered for publication in Nature Communications.

The authors would like to thank the reviewers for their time, expertise, and dedication in reviewing our manuscript, please find below a detailed point-by-point response to the reviewers' comments and recommendations. **Revisions are highlighted in yellow in the revised manuscript and Supplementary Information (SI).**

Reviewer #1:

Electroactive polymers with high ferroelectricity and large electrostriction are highly desirable for flexible devices. The authors, however, have presented a truly unique strategy in this work. They have explored high-performance PVDF-based materials not through the conventional optimization of the crystal phase or the adjustment of the carbon chain structure, but through the synergism of the crystal and amorphous phases. This novel approach, coupled with 3D printing to introduce a new structure, has fully exploited the benefits of crystal grains throughout the PVDF film. The measured electrostrictive coefficient and piezoelectric constant are notably higher than the values reported in the literature, indicating a significant advancement in this field. This work also provides fresh insight into the correlation between dielectric, electrostrictive, and ferroelectric properties in electroactive polymers. Therefore, I highly recommend it for publication in Nature Communications once the following issues are addressed.

Response: The authors acknowledge the constructive comments from the reviewer, which encourage us to continue our passion in this area.

COMMENT 1#: The authors mention ultra-high ferroelectricity in the title and abstract. Please briefly introduce the techniques and principles used to enhance ferroelectricity in PVDF materials.

Response: We thank the reviewer for pointing out this issue. The ferroelectric properties of a material are closely related to its electrostrictive properties. Previous works on ferroelectric/piezoelectric properties of PVDF have usually been based on the improvement of β phase¹, though the optimized d_{33} is mainly lower than 35 pm/V. In recent years, there have been

reports on ferroelectric performance improvement through enhancing electrostriction and polarization ($d_{33}=-1050$ pm/V)², or inducing the morphotropic phase boundary (MPB) to improve charge transport capacity ($d_{33}=-63.5$ pm/V)³. Given these two important progresses, we believe that our BUE-PVDF will use high electrostriction and charge transport capacity to achieve high ferroelectricity for PVDF-based electro-mechanical polymers.

Action taken: We have added the following sentences on page 2 of the manuscript:

“In fact, the improvement of ferroelectric property in PVDF is generally based on enhancing the composition and polarization of its β phase^{2, 11}.”

Subsequently, on page 10 of the manuscript, we have introduced the strategies and methods to improve the ferroelectric performance of PVDF in detail as follows:

“Note that previous efforts to improve the piezoelectric performance in PVDF are all based on enhanced polarization and β crystal phase¹¹, mainly including the annealing method to enhance the β phase content^{50, 51}, the poling method to induce the β phase and polarization altogether^{52, 53, 54}, the stretching method to facilitate the phase transformation from α to β phase^{55, 56}, the electrospinning method to prepare piezoelectric nanofibers in in-situ stretching and poling at the same time^{57, 58}, and chemical modification to increase the TTTT conformation chain (which will lead to higher β fraction in its crystal phase) for PVDF^{59, 60}.”

COMMENT 2#: This article points out that increased pore density improves the dielectric properties of materials in Fig. 2E. Could it be possible that such improvement is due to the increased conductivity caused by the high porosity?

Response: Thanks to the reviewer for the constructive comments. We agree that increasing the number of large pores can significantly enhance the conductivity of PVDF. Jin *et al.* pointed out that a widened D-E curve with the same slope can be used to detect the increased conductivity and the risk of charge leakage⁴. To verify the effect of pore-induced conductivity on dielectric properties, we designed a batch of PVDF film samples (made by Kynar® 721 powder) with different porosities. As shown in Fig. R1A, the corresponding residual electrical displacement of the three samples with 12 wt.%, 14 wt.%, and 16 wt.% PVDF concentration are 0.0017 $\mu\text{C}/\text{cm}^2$,

0.0029 $\mu\text{C}/\text{cm}^2$, and 0.1476 $\mu\text{C}/\text{cm}^2$, respectively. Therefore, the D - E curve gradually widened with the increasing PVDF concentration. Since the applied electric field is always less than 6 MV/m, it is much lower than the threshold electric field required to activate the β phase⁵. Therefore, the widening phenomenon in Fig. R1A should be due to the enhancement of the overall conductivity.

To further investigate this widening phenomenon, we compare the porosity structure of the samples with distinct residual electrical displacement. For the samples with less residual electrical displacement (12-14 wt.% PVDF concentration) in Fig. R1A, we cannot find any large pores (Fig. R1B), and the size of the existing pores is usually less than 2 μm (Fig. R1C). In contrast, for the sample with the most widened D - E curve (16 wt.% PVDF concentration) in Fig. R1(A), there are many macroscopic pores with a size larger than 50 μm widely distributed in the gap area (Fig. R1D and R1E). This significant difference in porosity structure between Fig. R1C and Fig. R1E is highly likely due to the less fluent of the printed droplet at a raised PVDF concentration. During solidification, the droplet with higher concentration becomes more reluctant to flow from the track area to the gap area. Thus, the shortage of material at the gap area may tend to form macroscopic pores or even through holes, resulting in an apparent widening of the D - E curves.

Nevertheless, the relative permittivity of all samples in Fig. R1A is around 14 (the slope of the gray region) despite their significant differences in conductivity. Their similarity in permittivity proves that the change in the overall conductivity does not influence the dielectric properties of BUE-PVDF. In this work, the enhanced permittivity of BUE-PVDF is achieved by the high density of small and medium pores (Fig. 2E) with their size smaller than 10 μm (Fig. S7). Considering the much smaller pore size and narrower D - E curve than that in the 16 wt.% sample in Fig. R1A, we believe that the increase of dielectric properties in BUE-PVDF should not stem from the pore-induced extra conductivity.

Action taken: We have added the whole discussion on this issue into Supplementary Note 5 and added Fig. R1 as Supplementary Fig. S4 to analyze the relationship between the charge leakage and the dielectric performance of the material in detail.

Fig. R1. (A) The D - E curve and dielectric results of samples with different ink concentrations and conductivities. (B)-(C) The SEM patterns exhibiting the topological and porosity structure in a sample at 13 wt.% with different magnifications. (D)-(E) The SEM patterns exhibiting the topological and porosity structure in a sample at 16 wt.% with different magnifications.

COMMENT 3#: In Fig. 1, the authors classify the porosity structures as gap and track area but never mention this classification again in the following characterization results (e.g., Fig. 2E). An explanation is suggested.

Response: The gap and track area in Fig. 1 distinguish the porosity structure in regions with different curing rates (Fig. 1A) from a macroscopic perspective. Generally, the PVDF ink in the track area is denser and thus leaves smaller pores with less density after curing (Fig. S7A). However, the porosity structure is also affected by many other factors. For instance, plenty of medium pores are observed in the track area of the samples prepared by a mixed PVDF powder (Fig. S7D and S7G). Therefore, the classification of porosity structure based on the printing region does not necessarily guarantee the same type and distribution of pores. Given that the current work mainly focuses on the microstructure's effect on Electro-mechanical (EM) performance in PVDF, this classification is no longer suitable for analyzing the relation between porosity structure and the EM properties. Therefore, we begin classifying the porosity structures through their pore size in the following discussion.

COMMENT 4#: Since Fig. 2F and 2G demonstrate a high inverse piezoelectric effect, please explain why the corresponding hysteresis curves are not similar to those of the typical ferroelectric material.

Response: Fig. R2 compares the hysteresis curves of commercial PVDF and BUE-PVDF under different applied cyclic electric fields. Typically, the β phase in PVDF can only be activated when the amplitude of the applied electric field exceeds a particular threshold value⁵. For instance, when the electric field amplitude exceeds 88 MV/m (Fig. R2A), the β phase of commercial PVDF is activated, and its hysteresis curve exhibits a typical ferroelectric style: the sample is polarized and maintains a high remanent polarization as the electric field becomes zero. In contrast, when the amplitude of the applied electric field is less than 60 MV/m (Fig. R2B), the β phase of commercial PVDF cannot be effectively activated. At this time, the electrical displacement D of the sample is approximately linearly increased with the applied electric field E ^{4,6}, which is similar to the hysteresis curve of dielectric materials.

In contrast, this work significantly reduces the threshold electric field (coercive field) to activate the β phase in the BUE-PVDF sample to less than 40 MV/m (Fig. R2C). This large discrepancy in threshold electric field between commercial PVDF and BUE-PVDF should probably related to the electro-lubricating effect. With the printed charge stored throughout the film, the BUE-PVDF successfully minimized the required applied field to achieve a $7 \mu\text{C}/\text{cm}^2$ electrical displacement from 112 MV/m in Fig. R2A to 46 MV/m in Fig. R2C. In addition, the BUE-PVDFs exhibit much higher permittivity in Fig. R2D (29.2 on average) than normal PVDF, which is 2.5 times higher than the average relative permittivity of commercial PVDF in Fig. R2B.

Action taken: We have added the discussion on this issue into Supplementary Note 7 and added Fig. R2 as Supplementary Fig. S5 to further explain the change on the D - E curve for the BUE-PVDF.

Fig. R2. The polarization hysteresis curves of a commercial β -phase PVDF (A) at high electric field and (B) at low electric field, respectively. The polarization hysteresis curves of a BUE-PVDF film (C) at high electric field and (D) at low electric field, respectively.

COMMENT 5#: In Fig. 4B, we find a typical parabolic (electrical field vs. strain) curve from BUE-PVDF, while the curve changes linearly in Fig.4C. What factors result in such a difference?

Response: It should be noted that both the PFM test methods in Fig. 4B^{7, 8} and the photonic displacement sensor method⁴ in Fig. 4C accurately reflect the sample's strain with the varied electric field. Their different curve shapes are due to the different polarization within the samples. Theoretically, the electro-deformation S_{33} of an electrostrictive material depends mainly on the applied electric field strength E and the remanent polarization P_r ⁵ as shown below:

$$\begin{aligned}
S_{33} &= Q_{33}D^2 = Q_{33}(\varepsilon_r\varepsilon_0E + P_r)^2 \\
&= 2Q_{33}\varepsilon_r\varepsilon_0P_rE + Q_{33}P_r^2 + Q_{33}\varepsilon_r^2\varepsilon_0^2E^2.
\end{aligned} \tag{R1}$$

When the applied electric field increment is dE , the corresponding electro-deformation increment dS_{33} is:

$$\begin{aligned}
dS_{33} &= 2Q_{33}\varepsilon_r\varepsilon_0P_r dE + 2Q_{33}\varepsilon_r^2\varepsilon_0^2E dE \\
&= 2Q_{33}\varepsilon_r\varepsilon_0(P_r + \varepsilon_r\varepsilon_0E) dE.
\end{aligned} \tag{R2}$$

In Fig. 4B, we adopted a PFM-based deformation test method. Each data point of the curve (E_i, S_i) is the peak strain S_i corresponding to a sine wave in the $0-E_i$ range. The remanent polarization $P_{r,i}$ corresponding to each data point increases with E_i . Therefore, according to Eq.(R2), dS_{33} increases approximately linearly with E , and the overall S_{33} increases approximately parabolically with E . If we select E_i near point 0 in Fig. 4B, the corresponding $P_{r,i}$ is also close to 0, so the S - E curve is close to a standard electrostrictive parabola.

In the ferroelectric analysis (Fig. 4C), the strain curve is a real-time strain measured by optical methods. In this method, we applied a sinusoidal electric field with a fixed magnitude for cycling loadings. Since the remanent polarization P_r of the sample is determined by the magnitude of the electric field, the P_r at each point in the curve almost remains constant. Considering that the dS_{33} of the BUE-PVDF curve in Fig. 4C decreases with increasing E , P_r should be in the opposite direction to E . So, when $E \rightarrow 0, -P_r \gg \varepsilon_r\varepsilon_0E$. In this case, the slope dS_{33}/dE is approximately constant, that is, the S_{33} increases linearly with E under the condition of low electric field.

Action taken: This analysis has been included in Supplementary Note 15 to understand better the connections and differences between the PFM test method and the photonic displacement sensor method.

COMMENT 6#: In Fig. 4C, please include a more detailed discussion about other PVDF-based materials to compare the strategies used to enhance ferroelectricity.

Response: Thanks to the reviewer for the helpful reminder. We agree that developing a more detailed discussion concerning ferroelectric optimization strategies based on the referred data in Fig. 4C is necessary. Therefore, related information has been added in the manuscript.

Action taken: As mentioned above (R1C1), we have added the following text on page 11 of the manuscript: “*Note that previous efforts to improve the piezoelectric performance in PVDF are all based on enhanced polarization and β crystal phase¹¹, mainly including the annealing method to enhance the β phase content^{50, 51}, the poling method to induce the β phase and polarization altogether^{52, 53, 54}, the stretching method to facilitate the phase transformation from α to β phase^{55, 56}, the electrospinning method to prepare piezoelectric nanofibers in in-situ stretching and poling at the same time^{57, 58}, and chemical modification to increase the TTTT conformation chain (which will lead to higher β fraction in its crystal phase) for PVDF^{59, 60}.*” to introduce the current promoting strategies in detail.

Reviewer #2:

This manuscript describes the fabrication of 3D electret structure consisting of PVDF via electro hydrodynamic printing to attain maximum electrostriction effects. Various samples with different ink concentrations, printing times, voltages and thicknesses were experimented. Through surface printing of charges utilizing the bottom-up technique, the authors could obtain a maximum electrostrictive coefficient value of $33.324 \times 10^{-18} \text{ m}^2/\text{V}^2$ which is 104 times greater and attained a piezoelectric constant d_{33} of 298 pm/V which is close to 10 times than the commercial PVDF films, making them suitable for use in smart devices and systems.

Response: The authors acknowledge the constructive comments from the reviewer, which encourage us to continue our passion in this area.

COMMENT 1#: The authors should cite the previously reported electrostrictive coefficients observed for PVDF-based samples obtained through surface modification and electron irradiation for effective comparison with the printed sample.

Response: Thanks to the reviewer for the constructive comment. We agree that the specific literature citations on improving the electrostrictive properties of PVDF samples through surface modification and other methods should be given in the introduction part. Therefore, related information has been added in the manuscript.

Action taken: We have added more literature on page 2 of the manuscript. To prevent the weakening of the primary narrative logic in the introduction, we choose to include the elaboration of this issue in Supplementary Note 16 to introduce the previous methods to promote the electrostriction for PVDF-based materials through surface modification and electron irradiation. Moreover, several significant progresses in the electromotive properties of PVDF-based material and their comparison with the printed PVDF are also exhibited in Fig. 4A.

COMMENT 2#: On which substrate was the EHD printing carried out before placing it onto the glass substrates?

Response: Fig. R3 presents the position of the sample, substrate, and needle during the printing process. A grounded metal as the negative electrode is underneath the glass substrate, and the needle is connected to the power supply as the positive electrode. Meanwhile, an insulating ceramic plate is installed below the metal plate to control the curing temperature. During the printing process, a tiny ink droplet is deposited on the glass substrate and solidified layer by layer to form the whole film. After curing, we will remove the PVDF film sample from the glass substrate for structural characterization or further dielectric and ferroelectric analysis. Therefore, throughout the EHD printing process, we always use the glass substrate as the collector of the ink droplet.

Action taken: We have added Fig. R3 as Supplementary Fig. S1 to clearly show the setup of the EHD printing system in this study.

Fig. R3. (A) Scheme of the EHD printing system during the printing process. (B) Photograph showing the setup of the substrate and the electrode in the EHD printing system.

COMMENT 3#: Though ferroelectric studies were carried out to better reflect dielectric properties, whether the authors tried to compare the results by measuring the same through impedance analysis?

Response: Theoretically, the impedance method measures the dielectric properties in different frequency ranges at low electric fields, while the ferroelectric method measures the average dielectric property through different electric fields at the same frequency^{4, 9}. To verify the difference between these two methods in dielectric measurements, we added the measuring results by impedance analysis for all samples in Fig. 2E. As shown in Fig. R4A, both methods provide consistent measuring results for the samples with various PVDF concentrations. However, the points from ferroelectric analysis are always higher than their impedance analysis counterparts. Further comparison among these samples exhibits a proportional relationship between the dielectric measurement results of the two methods in Fig. R4B. There could be two reasons for the 40% higher value obtained from the ferroelectric method. Firstly, the measurement frequency of the impedance method is 100 Hz, which is much higher than the 10 Hz measurement frequency for the ferroelectric method. Thus, the increase in frequency results in a lower dielectric value¹⁰. Secondly, the applied voltage of the impedance method is only 0.5V, which is much lower than the voltage range of the ferroelectric method (0-100 V). Lower voltages also have an impact on dielectric results⁶.

Action taken: We have added the following text on page 13 of the manuscript: “*The dielectric property (relative permittivity) can also be measured from the slope of the D-E curve in ferroelectric analysis. Note that all the dielectric values in this work are measured using the ferroelectric method, except the permittivity values obtained in Fig. 3C.*” to make it clear. And we have also added the whole discussion into the SM as the Supplementary Note 8 and added Fig. R4 into the SM as Supplementary S6 to provide complete information about this issue.

Fig. R4. (A) Variation of relative permittivity with PVDF concentration by ferroelectric method and impedance method respectively. (B) Comparison of different methods by linear fitting.

COMMENT 4#: Reason for specifically choosing 534000 g/mol and 275000 g/mol molecular weight of PVDF is not substantiated. How the ink concentrations were fixed for all samples?

Response: PVDF powder with 275,000 g/mol molecular weight (Kynar® 721) is widely used as raw material due to its wide range of sources, low price, and stable performance^{11,12}. However, the dielectric property of film entirely made by Kynar® 721 powder is merely increased with the adjustment of ink concentration (Fig. R5). Therefore, we attempt to mix Kynar® 721 powder with a higher molecular weight of 534,000 g/mol (Kynar® 761) to better adjust the internal porosity structure (Supplementary Fig. S7). Consequently, we can successfully present a strong effect of ink concentration on the dielectric properties of BUE-PVDF in Fig. 2E.

In the mixing process, Kynar® 721 powder and Kynar® 761 powder with the same weight is well mixed. Then, the corresponding solvent is added to the mixed powder to obtain specific concentrations of PVDF ink solutions with a mixing weight ratio of 1:1.

Action taken: We have provided a detailed information about this issue in Supplementary Note 3.

COMMENT 5#: Can the authors contrast the effect of PVDF blending in sample 5 with other samples (1-4)?

Response: Fig. R5A presents the dielectric properties of the samples made by single powder ($M_w=275,000$ g/mol) in their printable concentration range. Note that this series of samples is identical to those shown in Fig. R1. The relative permittivity values of the single powder BUE-PVDF are ranged between 12.5-17.5, while that values from their mixed powder counterparts are ranged between 18-36. These limited dielectric values in single powder BUE-PVDF could be explained by their porosity structures in Fig. R5D-R5E. Due to their similar size in PVDF particles, the pore sizes of single powder samples are much lower than their mixed powder counterparts as shown in Supplementary Fig. S7. Therefore, the much lower density of small and medium pores in single powder samples also leads to a lower permittivity. Meanwhile, since the single powder PVDF becomes stickier and less fluent with an increased concentration, the pore size of the 12 wt.% sample (Fig. R5B and R5C) is higher than that of the 15 wt.% sample (Fig. R5D and R5E). Also, the shortage of ink material at the gap area (Fig. R5E) in a high-concentration sample may tend to form large pores and even macroscopic pores (Fig. R1E). In contrast, the PVDF ink with mixed powder tends to produce small pores during solidification due to the significant difference in size between the two PVDF powders. Therefore, the mixed sample can achieve a higher and broader range of permittivity value with many small and medium pores adjusted by the ink concentrations (Fig. S7). Consequently, the ink with mixed powder is adopted in Fig. 2E to demonstrate better control through single printing parameter, i.e., PVDF concentration on the dielectric property of BUE-PVDF.

Action taken: This discussion is included in Supplementary Information Note 3 and Fig. R5 is added as Supplementary Fig. S2 to explain the use of mixed PVDF powders in detail.

Fig. R5. (A) The relationship between dielectric values and the ink concentrations for samples prepared by PVDF ink with single powders and mixed powders over their printable concentration range, respectively. (B) and (C) The porosity structure in PVDF sample made by mixed powders at 12 wt.% in its track area and gap area, respectively. (D) and (E) The SEM image exhibiting the porosity structure in PVDF sample made by single powder at 15 wt.% in its track area and gap area, respectively.

COMMENT 6#: In general porous structure leads to charge loss, how efficient the printed electrode?

Response: As Li *et al.* noted¹³, the electret charged through a macroscopic porous structure will bring the risk of charge loss. This phenomenon has also been found in Fig. R1: the risk of severe leakage, i.e., the dramatic widening of the hysteresis curve in Fig. R1A, is due to macroscopic pores in Fig. R1E. Therefore, the charge loss may occur in BUE-PVDF like ordinary electret materials. However, ordinary electrets rely on macroscopic pores to inject charge, so their leakage is unavoidable. In contrast, BUE-PVDF mainly relies on small pores to preserve a high density of charge during the printing process (Supplementary Fig. S8), thus the porosity structure in BUE-PVDF will not necessarily lead to charge loss.

Meanwhile, Fig. R1A proves that the dielectric properties in BUE-PVDF do not considerably increase with the number of macropores and the extent of leakage. In other words, the improved dielectric properties in BUE-PVDF (Fig. 2E) are not due to the macroscopic pore structure (Fig.

S7). The risk of leakage in BUE-PVDF can be avoided by carefully controlling the pores' size by adjusting printing parameters like ink concentration (Fig. R1, Fig. R5).

Action taken: We have discussed similar issue in second comment from the first reviewer (R1C2) and have added the whole discussion on this issue into Supplementary Note 5 and added Fig. R1 as Supplementary Fig. S4 to analyze the relationship between the charge leakage and the dielectric performance of the material in detail.

COMMENT 7#: Any changes in the crystallinity noticed from KPFM studies between BUE-PVDF and Solvent-casted PVDF which may have a direct effect on the surface activity?

Response: Fig. R6 summarizes the relationship between crystallinity and surface potential in both groups of samples (H1-H7 by printing, L1-L3 by solvent-casting). As the crystallinity increases from 49.4% to 64.1% for casted samples, the surface potentials are around -350 mV. The crystallinity distribution of H1-H7 is from 56.08% to 67.53%, and the corresponding surface potential ranges from -700 mV to -1100 mV. Generally, the surface potential of the casted sample is significantly higher than that of the printed sample under the same crystallinity. Such a lower surface potential in BUE-PVDF should stem from the accumulation of negative charges on the surface of the BUE structure. As shown in the manuscript Fig. 1A, the interior accumulation of positive charge causes the film's surface to adsorb negative charges, thus decreasing surface potential.

At the same time, we found that the surface potential of the casted sample hardly changed with the increase of crystallinity. In contrast, when the crystallinity of the printed sample exceeds 62%, its surface potential decreases significantly with the increase in crystallinity. This tendency shows the different sensitivity of the surface activity on crystallinity between the casted and printed samples.

To clarify the origin of the enhanced permittivity, we also present the relationship between these samples' crystallinity and permittivity in Fig. R6B. The permittivity values of both printed and casted samples increase with the crystallinity. However, when the crystallinity of the printed sample is higher than 64%, the slope of the permittivity with the change in crystallinity increases

rapidly. Considering that the stored charge of the corresponding sample in Fig. R6A substantially increases with crystallinity, we believe that the rapid increase for permittivity after 64% crystallinity in Fig. R6B should come from the reduced surface potential. In other words, the charge density of the BUE structure influences the dielectric properties more effectively than the crystallinity. Therefore, by varying its stored charge density, BUE-PVDF can achieve a broader range of dielectric properties in Fig. 4A than the casted or untreated sample.

Action taken: This discussion on the relationship among the dielectric property, the surface potential and the crystallinity in the printed and solvent-casted PVDF are added in Supplementary Note 13, and Fig. R6 is added as Supplementary Fig. S11 to further analyze the difference in the major factors for the dielectric property in the printed and solvent-casted PVDF respectively.

Fig. R6: (A) The variation of surface potential of BUE-PVDF (black square) and casted PVDF (red circle) with different crystallinity, respectively. (B) The variation of relative permittivity of BUE-PVDF (black square) and casted PVDF (red circle) with different crystallinity, respectively.

COMMENT 8#: According to Fig 2c, as relative permittivity is greater for 12 wt % of PVDF printed sample, charge density is greater for that sample. Whether the surface potential measurements were done for the corresponding sample to observe its pore density?

Response: Thanks for the excellent suggestion. This superb question can lead us to think more deeply about this type of BUE material. This paper focuses on enhancing the electro-mechanical properties of the BUE-PVDF materials with a unique charge structure. To further improve and

understand this type of material, it is necessary to develop the corresponding characterization method of this BUE structure. However, it is not easy to directly observe the interior charge of the material. Thus, we have found a method to indirectly characterize the charge distribution inside the film by measuring dielectric and surface potential.

As shown in Fig. R7A, the porosity structure of the high permittivity sample is relatively simple compared to the medium permittivity sample in Fig. S8. We observed a significant decrease of about 1.5 V near the pore. The concentration of negative charges around the pore agrees well with the phenomenon observed in Fig. S8. This result indicates that the pores can store charges in both medium and high permittivity samples. In other words, the concentration of negative charges near the smaller and medium pores is commonly observed (Fig. S8, Fig. R7D) in BUE-PVDF samples regardless of their permittivity.

Considering the larger pore size ($>10\mu\text{m}$) and pore distance in high permittivity samples, we need to observe the charge distribution with a more extensive range ($40\times 40\ \mu\text{m}$, Fig. R7B). However, the surface potential does not decrease in Fig. R7E; it increases around these larger pores. This opposite phenomenon is probably related to the depth of the surface pore. When the depth of the pores is deep enough ($>2\ \mu\text{m}$), the positive charge at the interior would transport through the deep pores and neutralize with the negative charge on surface. Thus, the surface potential distribution around the larger pore (Fig. R7B) becomes much higher than that around their smaller pore counterpart (Fig. R7A).

To further clarify the character of large pores, we also observed the surface potential distribution in casted samples. As shown in Fig. R7C, there are large pores similar to the high permittivity sample in Fig. R7B. However, the charges do not tend to accumulate in the vicinity of the pore (Fig. R7F), and the surface potential fluctuates only about 100 mV over the entire observation area. Unlike the printed samples, casted samples do not store positive charges during the fabrication process. As a result, the large pores on the surface have difficulty absorbing negative charges (Fig. R7D) and lack internal positive charges to increase their surface potential. (Fig. R7E).

Generally, we use KPFM to characterize the surface potential and understand BUE-PVDF's internal charge distribution. Since the charge storage capacity in BUE-PVDF originates from the charge accumulation at the pores, the estimation of charge accumulation capacity on different

pore types is crucial. With the pore size in Fig. 3D and Fig. S8 less than 1-2 μm and pore depth less than 300 nm, the charge accumulation capacity of these small and medium pores can be well characterized by the KPFM method. However, for those high-permittivity samples in Fig. 2C and Fig. S7, the size of the pores responsible for the high permittivity (Figure S5I) is about 3-10 μm . Thus, their pore depth is likely to exceed the thickness of the monolayer film (2 μm), resulting in the neutralization of the internal positive charges with the surface negative charges (Fig. R7E). At this point, the measured surface potential from KPFM cannot reflect the condition of internal charge density. Therefore, finding an alternative method to estimate the internal charge distribution for high permittivity samples with larger pores is necessary in future studies.

Action taken: We have added this content as Supplementary Note 11 and have added Fig. R7 as Supplementary Fig. S9 to completely discuss the relationship between the surface potential and the dielectric property in BUE-PVDF.

Fig. R7. The comparison diagram between the topography image and potential distribution images for the 12 wt.% PVDF printed high-permittivity sample and casted sample. The topography image on the surface of BUE-PVDF film and the corresponding potential distribution image for (A and D) in the medium pore region of the printed sample, (B and E) in the large pore region of the printed sample, and (C and F) in the pore region of casted sample, respectively.

COMMENT 9#: From Fig. 3F, it is seen that medium sized pores are only agglomerated on the surface and near surface, whereas low density pores (blue) are evenly distributed in the interior of the samples. How do the authors correlate the charge density and pore density based on this?

Response: We're sorry for the unclear display of the porosity structure at the interior in Fig. 3F. Since the entire film thickness in Fig. 3F is around 75-81 μm , the 3D reconstructed topography with a thickness of 28 μm only exhibits a partial region inside the film. It is found that the pore size and pore distribution from the interior region (Fig. 3F) and surface region (Fig. S7 and Fig. S8) are similar. Therefore, we believe that the charge distribution of the PVDF-BUE sample from the surface and interior is also comparable. Nevertheless, we acknowledge that Fig. 3F does not adequately reflect the porosity structure inside the reconstructed region.

Action taken: Therefore, we add a video (SM video 1) showing the cross-sections of the reconstructed region along the thickness direction of the sample in Fig. 3F. From this video, we find that medium pores appear randomly on each slide. Therefore, we believe the medium pores do not agglomerate on the surface or near-surface region. We also have added text on page 9 in the manuscript as follow: “...from the internal region of a BUE-PVDF film with about 80 μm in thickness. The cross-sections of the reconstructed region along the thickness direction in Fig. 3F are given in Supplementary Video 1 to further exhibit the pore distribution.” to make a more thorough clarification on this issue.

COMMENT 10#: Graphs pertaining to strain behaviour as a function of electric field, electrostrictive coefficients and absolute β fraction should be provided.

Response: The BUE-PVDF sample in Fig. 4B is in corresponding to Fig. 2F, which is also sample 6-1 (Table S5) mentioned in Supplementary Information ($\epsilon_r=8.8$, $M_{33}=98.6\times 10^{-18}$ m^2/V^2 , $\beta\%=27.8\%$). The BUE-PVDF sample in Fig. 4C is the sample corresponding to Fig. 2G, that is, also the sample 6-7 in Supplementary Information ($\epsilon_r=56$, $M_{33}=324\times 10^{-18}$ m^2/V^2 , $\beta\%=23.4\%$). From the above results, we can see that the β phase contents of the two samples are not very high. In addition, the β phase content (23.4%) of the sample with higher dielectric and EM

performances in Fig. 2G is unexpectedly lower than that (27.8%) of the sample with inferior performances in Fig. 2F. Thus, the improvement of EM performance in BUE-PVDF does not depend on the enhancement of β phase.

To accurately estimate the composition of the β crystal phase, we combine two characterization methods: differential scanning calorimetry (DSC) and infrared spectra analysis (IR). With the help of DSC, we can precisely determine the overall crystallinity of the material. With the help of IR, we can obtain the proportion of the TTTT configuration (β phase) within the material in all configurations, i.e., the relative content of the β phase. By multiplying these two values, we can estimate the absolute content of the β phase with high accuracy. The DSC curves for these two samples with relative permittivity of 8.8 and 56 are shown in Fig. R8A. It shows that the lower permittivity (8.8) sample has a larger exothermic peak area, i.e., a higher degree of crystallinity. Meanwhile, The IR data measurements showed that the relative β contents of the lower and higher permittivity samples are $40.61 \pm 0.45\%$ and $40.64 \pm 0.76\%$, respectively. Therefore, the relative β content is approximately the same among these two samples. In summary, the absolute content of the β phase in the 8.8 and 56 permittivity samples is 27.8% and 23.4%, respectively. In Fig. R8B, we further compare the relationship between the dielectric properties and the β phase content among the samples in Fig. 2E. This figure suggests that the absolute content of the β phase remained almost unchanged for a series of samples with significantly varying pore structure and dielectric properties. This figure further proves that the traditional β -phase optimization is not the main reason for the ultra-high electrostriction and ferroelectricity in BUE-PVDF.

Action taken: We have made a revision on Fig. 2E in the manuscript based on the Fig. R8. Meanwhile, we have added text on page 6 in the manuscript as follow: “*We should highlight that the β phase fraction is almost constant with varying concentration, indicating that the performance improvement in BUE-PVDF does not originate from the traditional β phase enhancement.*” to better exhibit the relationship among the electrostriction, the dielectric property and the β phase fraction in the BUE-PVDF.

Fig. R8. (A) The DSC result of the samples with lower permittivity ($\epsilon_r=8.8$, black curve) and higher permittivity ($\epsilon_r=56$, red curve) respectively, (B) The relative permittivity value and the β phase fraction of the samples prepared with varied PVDF ink concentrations.

COMMENT 11#: What was the optimized PVDF wt % for relative permittivity value of 56?

Response: The sample series 6 in the supplementary information corresponds to the 7 data points in Fig. 2H and are listed in Table R1. We are sorry that we omitted the processing parameters of the 6th data point in the Supplementary Table S5. Here, we have added the information about this sample as sample 6-6. The sample with a relative permittivity of 56 in Fig. 2H corresponds to the current sample 6-7 in Table S5 (previously marked as samples 6-6) with a concentration of 10%. It should be noted that the microstructure and EM properties of PVDF films are affected by a variety of printing parameters such as substrate temperature, needle size, relative humidity, etc... PVDF ink concentration is only one of the relatively convenient parameters for regulating the microstructure and properties (Supplementary Fig. S7). In this study, we use this parameter as a single variable to adjust the pore structure and dielectric properties (Fig. 2E) accordingly. And successfully increase the relative permittivity of the film to 36. However, the effect of the printing parameters on the microstructure is coupled and very complicated. For instance, we believe that the further improvement of the relative permittivity in Fig. 2H is closely related to the effect of lower relative humidity (Supplementary Table S5) on the curing process of printing

droplets. Nevertheless, the influence of multiple printing parameters on pore structure and electro-mechanical properties needs to be further studied.

Action taken: We have made the correction on the Supplementary Table 5 as shown below.

Table R1. Printing parameters for the PVDF samples correspond to Fig. 2H.

Sample #	Relative permittivity	Solvent	Concentration (wt.%)	Substrate temperature (°C)	Relative humidity (%)
6-1	8.4	DMF	10	80	65
6-2	10.1	NMP	14	60	55
6-3	29.1	DMF	10	50	30
6-4	30.8	NMP	13	40	22
6-5	38.2	DMF	10	50	31
6-6	44.9	DMF	11	50	22
6-7	56.0	DMF	10	50	22

COMMENT 12#: What was the value of electrostrictive coefficient for solvent casted PVDF film?

Response: We prepared a set of solvent-casted samples and measured their electrostrictive performance using the ferroelectric method. The specific process is as follows: firstly, a certain weight (0.6507 g) of PVDF powder is prepared, and the corresponding weight (4.3556 g) of solvent NMP is added to obtain a 13 wt.% PVDF ink solution. Next, place the serum bottle of the solution in an ultrasonic cleaner at 40°C for 1h, take it out, and let it stand for a moment to remove air bubbles. Finally, the clear and transparent solution is dropped into a glass mold and spread out naturally. The casted samples were obtained after natural drying at room temperature. The measured relative permittivity is 11.3, $M_{33}=22.3\times 10^{-18} \text{ m}^2/\text{V}^2$ (Fig. R9A and R9C). From the SEM image of the casted sample in Fig. R9B, the distinct porous structure (porosity up to 23%, pore size generally above 10 microns) should result in the leakage phenomenon represented as the widening of the $D-E$ curve in Fig. R9A. This porous structure causes a decrease in Young's modulus and an increase in $M_{33}^{8, 14, 15}$. The results from Fig. R9A to R9C indicate that the solvent-cast method can also achieve high electrostriction due to its unique porosity structure.

However, the difficulty of curing process control may bring many bubbles, causing the film to become wrinkled and fragile. Hence, the poor workability and thickness control significantly limit the application prospects of the casted PVDF. In addition to the casted sample, we also measured the electrostrictive coefficient of the commercial sample (β phase, polarized, corresponding to the commercial PVDF in Fig. 4C) and found that its M_{33} value was around $8.54 \times 10^{-18} \text{ m}^2 \text{V}^{-2}$ (Fig. R9D), which is close to the results reported by Hughes¹⁶ ($10.7 \times 10^{-18} \text{ m}^2 \text{V}^{-2}$). However, this value is still much higher than $0.028 \times 10^{-18} \text{ m}^2 \text{V}^{-2}$ reported by Furukawa *et al.*¹⁷. This discrepancy may be related to the β phase optimization of the sample and the different degrees of their polarization.

Fig. R9. (A) The polarization hysteresis curves of the casted PVDF samples. (B) The SEM image of the porosity structure within the casted PVDF sample. (C) The S - E curve of the casted PVDF sample. (D) The S - E curve of the commercial-PVDF sample.

Action taken: We have added this content in the Supplementary Note 14 and have added Fig. R9 as Supplementary Fig.12 to compare these results with the BUE samples as well as the reported results in the literatures.

Reference

1. Zhang, L., Li, S., Zhu, Z., Rui, G., Du, B., *et al.* Recent progress on structure manipulation of poly (vinylidene fluoride) - based ferroelectric polymers for enhanced piezoelectricity and applications. *Advanced Functional Materials* **33**, 2301302 (2023).
2. Chen, X., Qin, H., Qian, X., Zhu, W., Li, B., *et al.* Relaxor ferroelectric polymer exhibits ultrahigh electromechanical coupling at low electric field. *Science* **375**, 1418-1422 (2022).
3. Liu, Y., Aziguli, H., Zhang, B., Xu, W., Lu, W., *et al.* Ferroelectric polymers exhibiting behaviour reminiscent of a morphotropic phase boundary. *Nature* **562**, 96-100 (2018).
4. Jin, L., Li, F., Zhang, S., Green, D. J. Decoding the Fingerprint of Ferroelectric Loops: Comprehension of the Material Properties and Structures. *Journal of the American Ceramic Society* **97**, 1-27 (2014).
5. Katsouras, I., Asadi, K., Li, M., Van Driel, T. B., Kjaer, K. S., *et al.* The negative piezoelectric effect of the ferroelectric polymer poly (vinylidene fluoride). *Nature materials* **15**, 78-84 (2016).
6. Furukawa, T., Nakajima, K., Koizumi, T., Date, M. Measurements of nonlinear dielectricity in ferroelectric polymers. *Japanese journal of applied physics* **26**, 1039 (1987).
7. Bonnell, D. A., Kalinin, S. V., Kholkin, A., Gruverman, A. Piezoresponse force microscopy: a window into electromechanical behavior at the nanoscale. *MRS bulletin* **34**, 648-657 (2009).
8. Nawaka, K., Putson, C. Enhanced electric field induced strain in electrostrictive polyurethane composites fibers with polyaniline (emeraldine salt) spider-web network. *Composites Science and Technology* **198**, 108293 (2020).
9. Mellinger, A. Dielectric Resonance Spectroscopy: a Versatile Tool in the Quest for Better Piezoelectric Polymers. *IEEE Transactions on Dielectrics & Electrical Insulation* **10**, 842 (2003).
10. Wentink Jr, T. Properties of Polyvinylidene Fluoride. I. Dielectric Measurements. *Journal of Applied Physics* **32**, 1063-1064 (1961).
11. Zheng, J., He, A., Li, J., Han, C. C. Polymorphism control of poly (vinylidene fluoride) through electrospinning. *Macromolecular rapid communications* **28**, 2159-2162 (2007).
12. Azzaz, C. M., Mattoso, L. H., Demarquette, N. R., Zednik, R. J. Polyvinylidene fluoride nanofibers obtained by electrospinning and blowspinning: Electrospinning enhances the piezoelectric β - phase—myth or reality? *Journal of Applied Polymer Science* **138**, 49959 (2021).
13. Li, X., Wang, Y., Xu, M., Shi, Y., Wang, H., *et al.* Polymer electrets and their applications. *Journal of applied polymer science* **138**, 50406 (2021).
14. Assagra, Y., Altafim, R., Carmo, J., Altafim, R., Gerhard, R. A New Route to Piezo-Polymer Transducers: 3D Printing of Polypropylene Ferroelectrets. *IEEE Transactions on Dielectrics and Electrical Insulation* **27**, 1668 (2020).

15. Liu, Q., Richard, C., Capsal, J. F. Control of crystal morphology and its effect on electromechanical performances of electrostrictive P(VDF-TrFE-CTFE) terpolymer. *European Polymer Journal* **91**, 46-60 (2017).
16. Hughes, O. R. Frequency dependence of hysteresis associated with the electromechanical performance of PVDF film. *Journal of Polymer Science Part B: Polymer Physics* **45**, 3207-3214 (2007).
17. Furukawa, T., Seo, N. Electrostriction as the origin of piezoelectricity in ferroelectric polymers. *Japanese Journal of Applied Physics* **29**, 675 (1990).

The authors would like to thank the reviewers for their time, expertise, and dedication in reviewing our manuscript, please find below a detailed point-by-point response to the reviewers' comments and recommendations. **Revisions are highlighted in green in the revised manuscript and Supplementary Information (SI).**

Reviewer #2:

Though authors have considerably answered all the queries, there are few other comments which warrants further clarification.

COMMENT 1#: The authors should better clarify comment no. 5, as the effect of blending two different molecular weight PVDF is not effectively contrasted with other samples. It would be better, if authors can compare in terms of performance improvement with other samples.

Response:

Fig. R1. The effect of powder blending in sample 5 series on performance improvement in comparison with other samples.

Fig. R1 exhibits the dielectric and electrostrictive properties comparison between the mixed

powder samples and other samples in the manuscript. Compared with the single powder samples (green triangle, Single P.), the range for the ϵ_r values of the mixed powder samples (blue circle, S5 series) is extended from 8-18 to 18-35, consistent with the results in Fig. S2. This tendency indicates that the dielectric properties of the mixed powder samples can be easily manipulated by adjusting their PVDF concentration. Therefore, we utilize the mixed powder samples in the manuscript (Fig. 2E) to demonstrate the relationship between porosity structure and dielectric properties with varying ink concentrations.

For the mixed powder samples, each sample's electrostriction coefficients (M_{33}) are generally increased with their permittivity values. This increasing slope is close to that of the final optimized samples (red pentagram, S6 series, also shown in Fig. 2H). Still, for other samples, such as S1 series samples with different voltages (light blue pentagons), S3 series samples with different thickness (purple diamonds), and the single powder samples with various concentrations (Single P.), the increasing slope is quite different. It may even reduce to zero or present as a negative value. More importantly, the electrostrictive performances of these samples are much lower than those of the final optimized samples (S6 series) with comparable permittivity. These results prove that the dielectric and electrostrictive properties are unlikely to be simultaneously optimized by controlling only one printing parameter. In other words, optimizing the overall electro-mechanical performances in BUE-PVDF should be governed by multiple printing parameters.

The insufficient optimization of the electrostrictive performance among these samples is probably due to the difference in charge density between the internal and surface regions of the printed film. In the current work, the M_{33} value of the sample is based on the measurement of the film's overall strain. In contrast, the measured relative permittivity of the sample only reflects dielectric property on the surface region (near the detecting electrode)^{1, 2}. Therefore, the electrostrictive property can be optimized once the localized charge density at the film's interior is close to its upper limit. This upper limit should be related to the dielectric property on the surface due to the comparable porosity structure throughout the whole film. Under such circumstances, the optimized electrostriction can be almost increased linearly with the permittivity in Fig. R1 (red shadowed area). Unfortunately, because of multiple factors^{3, 4, 5}, the printed charge may not be preserved in each layer at the same level during the printing process. Thus, the overall M_{33} values among these samples are usually lower than their theoretical upper limits due to the lower charge

density at the film's interior.

The testing results from more samples (black squares in Fig. R1) exhibit a common tendency: the printed sample with similar dielectric properties but prepared with different printing parameters may diverge widely in their electrostrictive performance. Meanwhile, the optimized electrostriction (the upper limits of M_{33} in Fig. R1) increases linearly with permittivity. In general, the electrostrictive performance of BUE-PVDF samples still improves with the enhancing permittivity. It is worth noting that the electro-mechanical (EM) properties for most BUE-PVDF samples are much higher than that of the casted or commercial PVDF sample (Fig. S12), demonstrating the effectiveness of BUE structure in improving the EM properties of PVDF.

Action taken: We have added the whole discussion on this issue into Supplementary Note 17 and added Fig. R1 as Supplementary Fig. S13 to compare mixed powder sample 5 series in terms of performance improvement with other samples.

COMMENT 2#: The optimized PVDF wt % for relative permittivity value of 56 in comment no.10 is also not clearly substantiated.

Response:

In this study, we propose that the electrical-lubricating effect from the stored charge near the pores improves the dielectric properties of BUE-PVDF. Thus, the permittivity value should be highly correlated with the porosity and pore distribution of the material (Fig. 2E and Fig. S7). Therefore, for the sample with $\epsilon_r=56$ in Fig. 2H (sample 6-7), its higher permittivity than other samples should originate from the further optimization of the porosity structure. As discussed in Comment 1, the best optimization of the porosity structure in this sample requires comprehensive control of various printing parameters such as printing speed, substrate temperature, air humidity, ink concentration, etc., rather than the variation of single parameters like ink concentration.

To illustrate the importance of porosity structure on these high-permittivity samples, we quantify their porosity structure parameters from SEM images through a unified measuring method. Fig. R2A and R2B show the mesh-like pattern of the printed samples 5-1 and 5-3 prepared from the mixed-powder ink. There are remarkable differences in localized porosity between the track and

the gap regions. Therefore, we must separately measure the porosity ρ_{track} , ρ_{gap} and area fraction Ar_{track} , Ar_{gap} in the track and the gap regions. Then, the overall porosity ρ of the samples can be expressed as:

$$\rho = Ar_{\text{track}}\rho_{\text{track}} + Ar_{\text{gap}}\rho_{\text{gap}} \quad (\text{R1})$$

Ar_{track} and Ar_{gap} 's evaluation relies on measuring each region's average length from SEM images with low magnification. As shown in Fig. R2A, we mark the boundaries of all track regions based on their distinct porosity and surface height. Then, the average length of each area (illustrated in Fig. R2A) along the X and Y directions, $L_{\text{track},x}$, $L_{\text{gap},x}$, and $L_{\text{track},y}$, $L_{\text{gap},y}$ are measured by the line intercept method. For each unit cell, the area fraction of each region can be approximately estimated as:

$$\begin{cases} Ar_{\text{gap}} = L_{\text{gap},x} \cdot L_{\text{gap},y} / \left[(L_{\text{track},x} + L_{\text{gap},x}) \cdot (L_{\text{track},y} + L_{\text{gap},y}) \right] \\ Ar_{\text{track}} = 1 - Ar_{\text{gap}} \end{cases} \quad (\text{R2})$$

Similarly, we reproduce samples 6-4 and 6-7 and calculate the area fraction of each region through the corresponding SEM images (Fig. R2C and R2D). These samples are prepared with the same printing parameters as shown in Table S5. The permittivity and electrostriction of these reproduced samples (6-4R and 6-7R) are closely comparable to those of the origin ones shown in Fig. 2H. It should be noted that we use more than four SEM images for each sample to ensure a reliable measurement of the area fraction.

ρ_{track} and ρ_{gap} can be directly measured from the average localized porosity from several high-magnified SEM images. An OpenCV (Open-Source Computer Vision Library) module in the Python library is adopted to identify the porosity in each SEM image. The original image is converted to a binary version with a careful threshold value setting. Hence, the localized porosity of the corresponding image is estimated as the ratio of black pixels (pore areas) to the entire image area. Fig. R2E-R2P show the porosity structure identification results with tolerable accuracy from the track and the gap region in the samples 5-3, 6-4R, and 6-7R, respectively. Similarly, we selected at least three SEM images for each sample to ensure a reliable estimation of the localized porosity. Suppose there are several different values of localized porosity in the same region type (e.g., track area in Fig. R2B). In that case, we will consider the frequencies of these localized porosity to further amend the overall porosity within the gap/track region.

Table R1 summarizes the porosity parameters measured from the mixed powder samples 5-1 and 5-3 and single powder samples 6-4R and 6-5R, respectively. It can be found that except for sample 5-1 with the lowest permittivity value (18), the localized porosity in the track regions of the other three samples (with permittivity value higher than 30) is greater than that in the gap region. Meanwhile, samples with higher localized porosity in the track region tend to have higher overall porosity and more excellent dielectric properties. This trend is also reflected in Fig S7A, D, G, and M. Since the formation of localized porosity structure in each region depends on the corresponding curing rates during the printing process, the different curing rates of the track and the gap regions make it unlikely to optimize the porosity structures of both regions simultaneously. Considering A_{track} (the area fraction of the track region) is usually more than 50%, the printing parameters that can optimize the track region's porosity structure will be more effective in improving the film's overall porosity and dielectric properties.

Note that the above analysis only examines the relationship between porosity structure and dielectric property in terms of the average porosity. However, the uniformity of the pore distribution (Fig. 3F), pore size, and pore density (Fig. S10) may also affect the dielectric properties. Compared to the mixed powder sample (Fig. R2B), the single powder sample (Fig. R2D) has a uniformly distributed pore structure within its track region (Fig. S7H, S7I, and Fig. R2G). Therefore, sample 6-7R can store more charges through more and smaller pores under similar porosity, thus significantly improve its dielectric performance.

Action taken: We have added the whole discussion on this issue into Supplementary Note 18 and added Fig. R2 as Supplementary Fig. S14, Table R1 as Supplementary Table S6 to further illustrate the importance of porosity structure on these high-permittivity samples.

Fig. R2. (A-D) The different distribution and size of porosity in the track and the gap regions of the printed PVDF samples corresponding to samples 5-1, 5-3, 6-4R, and 6-7R, respectively. (E-J) The SEM images and their correlated binary images for the samples' track region corresponding to samples 5-3, 6-4R, and 6-7R, respectively. (K-P) The SEM images and their correlated binary images for the sample's gap region corresponding to samples 5-3, 6-4R, and 6-7R, respectively.

Sample #	$L_{\text{track},x}$ (μm)	$L_{\text{track},y}$ (μm)	$L_{\text{gap},x}$ (μm)	$L_{\text{gap},y}$ (μm)	A_{rtrack}	ρ_{track} (%)	ρ_{gap} (%)	ρ (%)	ϵ_r	M_{33} (m^2/V^2)
5-1	117	130	222	225	58.5%	1.5	4.0	2.5	18	32
5-3	127	132	210	227	60.6%	7.3	4.3	6.1	36	120
6-4R	112	111	280	237	51.4%	5.1	2.7	3.9	32	250
6-7R	127	150	229	216	62.1%	10.2	3.7	7.7	59	336

Table. R1. The statistical parameters for the porosity structures in samples 5-1, 5-3, 6-4R, and 6-7R with different permittivity.

Reference

1. Jin, L., Li, F., Zhang, S. Decoding the fingerprint of ferroelectric loops: comprehension of the material properties and structures. *J. Am. Ceram. Soc.* **97**, 1-27 (2014).
2. Mellinger, A. Dielectric resonance spectroscopy: a versatile tool in the quest for better piezoelectric polymers. *IEEE Trans. Dielectr. Electr. Insul.* **10**, 842 (2003).
3. Mellinger A. Charge storage in electret polymers: mechanisms, characterization and applications. Universität Potsdam (2004).
4. Belhora F, Guyomar D, Mazroui MH, Hajjaji A, Boughaleb Y. Thickness effects of electret and polymer for energy harvesting: Case of CYTOP- CTLM and polyurethane. *Eur. Phys. J. Plus* **130**, 1-9 (2015).
5. Collins G, Federici J, Imura Y, Catalani LH. Charge generation, charge transport, and residual charge in the electrospinning of polymers: A review of issues and complications. *J. Appl. Phys.* **111**, 044701 (2012).